# Deciphering the Autoantibody Response to the OJ Antigenic Complex

**DOI:** 10.3390/diagnostics13010156

**Published:** 2023-01-03

**Authors:** Marvin J. Fritzler, Chelsea Bentow, Minoru Satoh, Neil McHugh, Anna Ghirardello, Michael Mahler

**Affiliations:** 1Cumming School of Medicine, University of Calgary, Calgary, AB T2N 1N4, Canada; 2Werfen Autoimmunity, San Diego, CA 92131, USA; 3Department of Human, Information and Science, University of Occupational and Environmental Health, Kitakyushu 807-8555, Japan; 4Department of Medicine, Kitakyushu Yahata-Higashi Hospital, Kitakyushu 805-0071, Japan; 5Department Pharmacy and Pharmacology, University of Bath, Bath BA2 7AY, UK; 6Unit of Rheumatology, Department of Medicine, University of Padova, 35122 Padua, Italy

**Keywords:** autoantibodies, OJ, myositis, antisynthetase syndrome

## Abstract

(1) Background: Myositis specific antibodies (MSA) are important diagnostic biomarkers. Among the rarest and most challenging MSA are anti-OJ antibodies which are associated with anti-synthetase syndrome (ASS). In contrast to the other tRNA synthetases that are targets of ASS autoantibodies (e.g Jo-1, PL-7, PL-12, EJ, KS, Zo), OJ represents a macromolecular complex with several ribonucleoprotein subunits. Therefore, the choice of the antigen in autoantibody assays can be challenging. (2) Methods: We collected two independent cohorts with anti-OJ antibodies, one based on a commercial line immunoassay (LIA) (n = 39), the second based on protein immunoprecipitation (IP) (n = 15). Samples were tested using a particle-based multi-analyte technology (PMAT) system that allows for the simultaneous detection of antibodies to various autoantigens. For the detection of anti-OJ antibodies, two different antigens were deployed (KARS, IARS) on PMAT. The reactivity to the two antigens KARS and IARS was analyzed individually and combined in a score (sum of the median fluorescence intensities). (3) Results: In the cohort selection based on LIA, 3/39 (7.7%) samples were positive for anti-KARS and 7/39 (17.9%) for anti-IARS and 14/39 (35.9%) when the two antigens were combined. In contrast, in samples selected by IP the sensitivity of anti-KARS was higher: 6/15 (40.0%) samples were positive for anti-KARS, 4/15 (26.7%) for anti-IARS and 12/15 (80.0%) for the combination of the two antigens. 18/39 (46.2%) of the LIA samples generated a cytoplasmic IIF pattern (compatible with anti-synthetase antibodies), but there was no association with the antibody levels, neither with LIA nor with PMAT. (4) Conclusions: The combination of IARS and KARS might represent a promising approach for the detection of anti-OJ antibodies on a fully automated platform.

## 1. Introduction

Myositis specific antibodies (MSA) represent not only important diagnostic biomarkers, but also help stratify myositis patients with particular clinical features, treatment responses and disease outcomes [1,2]. Although MSA, with the exception of anti-Jo-1, are not included in the most recent classification criteria for idiopathic inflammatory myopathies (IIM) [3,4,5], it has been reported that MSA as biomarkers for IIM outperform the current classification criteria [6]. Consequently, the standardization of autoantibody assays for the detection of MSA is of high relevance [7,8]. Although some MSA show a high degree of commutability between methods for their detection, others show greater variability [7,8,9]. Among the rarest and most challenging MSA are anti-OJ antibodies which are associated with anti-synthetase syndrome (ASS) and target the isoleucyl tRNA synthetase [10,11,12,13,14]. In contrast to the other tRNA synthetases that are targeted by autoantibodies (e.g., Jo-1, PL-7, PL-12, EJ, KS, Zo), OJ represents a macromolecular complex with several protein subunits [15]. Therefore, the choice of the antigen in autoantibody assays can be challenging and is critical for assay accuracy [15,16,17]. Although it is well appreciated that line immunoassays (LIA) lack accuracy for the detection of anti-OJ antibodies [18,19,20,21,22], they are widely used in clinical practice due to the ease of use and the lack of viable alternatives. Due to the cellular function and location of RNA synthetases, anti-OJ antibodies have been reported to generate a cytoplasmic indirect immunofluorescence (IIF) staining pattern on HEp-2 cells. In contrast to some of the other anti-synthetase antibodies (e.g., Jo-1, PL-7, PL-12), a fine pattern specificity has not been defined for anti-OJ antibodies by the International Consensus on ANA patterns (ICAP). In this context, it is important to point out that IIF on HEp-2 cells is insensitive for the detection of antibodies targeting a number cytoplasmic autoantigens [23]. Consequently, an alternative screening approach using lower sample dilutions has been proposed [23,24,25], but has not been widely adopted in conventional diagnostic laboratories. In addition, a novel fully automated particle-based multi-analyte technology (PMAT, Inova Diagnostics, research use only) has recently been developed for the detection of MSA [9,26,27,28]. However, due to the challenges of the anti-OJ autoantibody system outlined above, early versions of PMAT excluded this analyte [9,27,28]. In a large Japanese multi-center study, the clinical phenotype of anti-OJ positive myositis patients was summarized [10]. It appears that anti-OJ myopathy is strongly associated with interstitial lung disease (ILD) rather than myositis or Raynaud phenomenon (RP). Our study aimed to further decipher the autoantibody immune response to the OJ antigenic complex by means of immunoassays and epitope mapping with the ultimate goal of developing alternative and reliable methods for the detection of anti-OJ antibodies. 

## 2. Materials and Methods

Two independent cohorts of serum samples were included in this study: one collected based on anti-OJ antibodies identified by LIA (Myositis Profile 3: Euroimmun, Lübeck, Germany), and a second based on protein IP of radiolabeled cell extracts as previously reported [9]. The first cohort comprised of 39 anti-OJ positive samples were preselected on the results obtained using a commercial LIA (Myositis Profile 3: Mi-2, Ku, PM-Scl100, PM-Scl75, Jo-1, SRP, PL-7, PL-12, EJ, OJ, Ro-52/TRIM21, NT5c1A (Appendix A) and all positive samples based on the manufacturer’s suggested cutoff were included to preclude bias of the study by selecting only sample with high levels of reactivity. All samples were submitted for MSA testing based on the suspicion of IIM in the patients. Detailed clinical data for this cohort was not available. The cohort selected based on IP included a total of 15 samples [Bath Institute for Rheumatic Diseases (BIRD)/University of Bath serology service and UKMyoNet cohorts] as described previously [9]. A HEp-2 IIF assay (NOVA Lite HEp-2, Inova Diagnostics, San Diego, CA) was used to detect anti-cellular (AC) antibodies at a serum dilution of 1:80 and read on an automated instrument (NOVA View, Inova Diagnostics) which interpolated fluorescence intensity to an end point titer. AC IIF patterns were classified according to the International Consensus on Autoantibody Patterns (ICAP; https://anapatterns.org/trees-2021.php, (accessed on 20 September 2022)). Control or comparator serum samples were from health individuals (HI) and those with a diagnosis of inflammatory bowel disease (IBD); infectious disease (ID); rheumatoid arthritis (RA), or systemic lupus erythematosus (SLE).

### 2.1. Anti-OJ Autoantibody Testing

All samples were tested using the novel fully automated particle-based multi-analyte technology (PMAT) which utilizes paramagnetic particles with unique signatures and a digital interpretation system as described previously [9,26,27]. The PMAT technology (PMAT, Werfen, San Diego, CA, USA, research use only) allows for the simultaneous detection of antibodies to various autoantigens. For the detection of anti-OJ antibodies, two different antigens were deployed [lysyl tRNA Synthetase (KARS), isoleucyl tRNA Synthetase (IARS)]. The reactivity to the two antigens KARS and IARS was analyzed individually and combined in a score (sum of the median fluorescence intensities). For the combined score, the same median fluorescence intensity (MFI) threshold as the individual analytes was used.

In brief: recombinant full-length human IARS and KARS proteins were coupled to paramagnetic particles that carry unique signatures and incubated with diluted serum samples. After 9.5 min incubation at 37 °C, particles were washed and incubated 9.5 min at 37 °C with anti-human IgG conjugated to phycoerythrin (PE) to label the bound autoantibodies. After the final wash cycle, median fluorescence intensity (MFI) on the particles was captured using a digital imager and analyzed using proprietary algorithms to derive meaningful information for each analyte. 

### 2.2. Epitope Mapping

Epitope mapping was performed by PEPperPRINT (Heidelberg, Germany) as previously described [10]. The sequences of antigens IARS, MARS, DARS, EPRS, QARS, RARS, KARS, LARS, P38/AIMP2, P18/AIMP3 and P43/AIMP1 were linked and elongated with neutral GSGSGSG linkers at the C- and N-termini to avoid truncated peptides. The linked and elongated protein sequences were translated into 15 amino acid peptides with a peptide-peptide overlap of 13 amino acids. The resulting peptide microarrays contained 4130 different linear peptides printed in duplicate (8,260 peptide spots) and were framed by additional hemagglutinin (HA) (YPYDVPDYAG, 98 spots) and polio (KEVPALTAVETGAT, 96 spots) control peptides. Pre-staining of a custom peptide microarray utilized secondary and control antibodies in incubation buffer to investigate background interactions with the antigen-derived peptides that could interfere with the main assays. Subsequent incubation of other peptide microarray copies with the human serum pools at dilutions of 1:500 and 1:100 in incubation buffer was followed by staining with secondary and control antibodies. The serum pools included two positive pools (three anti-OJ positive sera each) and one anti-OJ negative control pool. Read-out was performed with a LI-COR Odyssey Imaging System at scanning intensities of 7/7 (red/green). The additional HA control peptides framing the custom peptide microarrays were simultaneously stained with the control antibody as internal quality control to confirm the assay quality and the peptide microarray integrity.

Quantification of spot intensities and peptide annotation were based on the 16-bit gray scale tiff files that exhibit a higher dynamic range than the 24-bit colorized tiff files. Microarray image analysis was done with PepSlide^®^ Analyzer and summarized in the Excel files listed in Material and Methods. A software algorithm interpolates fluorescence intensities of each spot into raw, foreground and background signals, and calculates averaged median foreground intensities and spot-to-spot deviations of spot duplicates. Based on averaged median foreground intensities, intensity maps were generated and interactions in the peptide maps highlighted by an intensity color code with red for high and white for low spot intensities. A maximum spot-to-spot deviation of 40% was tolerated, otherwise the corresponding intensity value was zeroed.

Averaged spot intensities of the assays with the human serum pools against the antigen sequences from the N-terminus of IARS to the C-terminus of P43/AIMP1 to visualize overall spot intensities and signal-to-noise ratios were also plotted. The intensity plots were correlated with peptide and intensity maps as well as with visual inspection of the microarray scans to identify the antibody responses of the human serum pools. In case it was not clear if a certain amino acid contributed to antibody binding, the corresponding letters were written in grey. For a better data overview, the baselines of the intensity plots were leveled.

Six synthetic soluble biotinylated peptides representing the key epitopes were synthesized and subsequently used in the novel PMAT immunoassay. Streptavidin coupled beads were used to immobilize the biotinylated peptides. 

### 2.3. OJ Antigens

Based on a recent publication [29], two proteins were selected and used for the current study: KARS1 (LysRS), Lysine-tRNA ligase Lysyl-tRNA synthetase Gene ID: 3735, Uniprot: Q15046, mRNA: D32053.1, 68.048 KDa 597 amino acids. IARS1 (IleRS), Isoleucine-tRNA ligase, Isoleucyl-tRNA synthetase, Gene ID: 3376, Uniprot: P41252, mRNA: U04953.1, 144.498 KDa, amino acids 1262 (see Figure 1; Appendix A).

### 2.4. Statistical Methods

Quantitative agreements were calculated using spearman correlation. Differences between groups were assessed using Mann–Whitney statistics. All statistics were done using Analyse-it software (Leeds, UK).

## 3. Results

### 3.1. Anti-OJ Positive Samples Identified by LIA vs. IP

First, we analyzed the LIA and IP sample cohorts. In the LIA cohort (n = 39), the median anti-OJ reactivity was 16 units with a minimum of 11 and a maximum of 86 units. In this cohort, 3/39 (7.7%) samples were positive for anti-KARS and 7/39 (17.9%) for anti-IARS as tested by PMAT. When the At a preliminary combined cut-off for the two peptides, 14/39 (35.9%) anti-OJ LIA positive samples tested positive for the combination of the two recombinant proteins using PMAT (results summarized in Table 1). The reactivity to the two antigens on PMAT was correlated (*rho* = 0.544, 0.37–0.74; *p* = 0.0003). When the results between PMAT and LIA were quantitatively compared, a trend was observed but the correlation was not significant. Based on the correlation in the reactivity we blasted the two antigens (IARS, KARS) for homology as an attempt to understand potential cross-reactivity (see Figure 1). Very low homology between the two antigens were observed.

Next, we analyzed the samples selected by IP and found 6/15 (40.0%) samples positive for anti-KARS and 4/15 (26.7%) for anti-IARS using PMAT. At a preliminary combined cut-off, 12/15 (80.0%) anti-OJ positive samples tested positive for the combination of the two antigens (Figure 2). 

When we compared the ratio of reactivity of anti-IARS vs. anti-KARS in the samples selected by LIA and IP, significant differences were observed (Figure 3). Samples selected based on LIA had significantly higher reactivity to IARS as compared to KARS (ratio 1.93 vs. 0.84, *p* = 0.003). 

### 3.2. Association of Anti-OJ Antibodies and Indirect Immunofluorescence (IIF) Staining Patterns

When the quantitative results obtained from LIA and PMAT were compared to the HEp-2 IIF results (Figure 4 and Figure 5; Appendix A), no clear association was observed between ANA positive and negative samples. More specifically, 18/39 of the LIA anti-OJ positive samples generated a cytoplasmic IIF pattern (compatible with anti-synthetase antibodies, AC-19, AC-20), but there was no association with the antibody levels on either the LIA or PMAT. The same was observed when analyzing the results based on the AC pattern following the ICAP classification. Lastly, we studied the association between reactivity to two OJ components, namely KARS and IARS, and the IIF pattern. For this comparison, we also did not observe a consistent association with a nuclear or cytoplasmic IIF staining pattern. However, a trend indicating that autoantibody levels (for all methods including LIA, PMAT-IARS, PMAT-KARS) were higher in samples with positive cytoplasmic staining, the difference did not reach statistical significance (data not shown).

### 3.3. Epitope Mapping

Epitope mapping revealed several immunodominant epitopes on components of the OJ complex (Figure 6). On a high level, the two pools with samples positive for anti-OJ antibodies showed higher reactivity when compared to the control pool which indicated the presence of linear epitopes. The human serum pool #1 showed the strongest and most complex antibody response. Reactive epitopes included the following sequences for pool 1: SGWEQ, TDQWERWWKNPEQ, LFQKLENDQ, KTSPKPAVVETVT (all MARS), ACPIVDSIE, VNEGLVD (all EPRS), TVGTSVG (KARS), SENVIQSTAVTTV, VVSGLVNHVPLEQ, DKELNPKKKIWEQ (all P43/AIMP1) and for pool 2: CGTDEYGTA (MARS), MLQPYMPTV (MARS), VRRDTGEKLTV (EPRS), HSSVKS (P38/AIMP2). Epitopes that exhibited very high reactivity included peptides derived from EPRS (ACPIVDSIE) and P43/AIMP1 (VVSGLVNHVPLEQ). To confirm the reactivity from the solid phase peptide array, 6 soluble biotinylated peptides were synthesized and tested with the serum pools used for discovery, as well as with individual serum samples. Although the reactivity was confirmed with the pool samples, low reactivity was found with the individual samples (data not shown). 

## 4. Discussion

### 4.1. Limitation of Current Assays for the Detection of Anti-OJ Antibodies

Although known for many years, anti-OJ antibodies remain challenging antibodies to detect [15,16,17]. Several studies have demonstrated that commonly used LIAs fail to reliably and accurately detect anti-OJ antibodies [16,18,19,20,21] (summarized in Table 2). More specifically, the study by Hamaguchi et al. reported that 0/9 anti-OJ positive samples as defined by IP were detected by LIA resulting in 0% sensitivity, although the specificity of the assay (100%) was high with 0/52 of controls testing positive. In contrast, Lackner et al. reported that all three anti-OJ positive patients had a diagnosis of IIM generating concerns about the specificity of the LIA [20]. Another study reported two anti-OJ positive patients as detected by LIA which were not confirmed by IP [22]. Last, Tansley et al. [16] compared the same LIA used in the previous studies in addition to a dot blot (vs. IP), and reported that both assays showed very low sensitivity in that the LIA failed to detect all 14 anti-OJ positive samples while the dot blot detected 1/14 as positive and another 2/14 as borderline. In a study by Cavazzana et al. 2/57 IIM samples tested positive for anti-OJ antibodies by IP, but none were positive by LIA. Vulsteke et al. studied MSA in 144 myositis patients and 240 controls which all tested negative for anti-OJ antibodies [30]. The lack of anti-OJ antibodies in this IIM group might be explained by the rarity of this antibody specificity but might also be related to the lack of sensitivity of LIA and dot blot. The data on the 240 controls confirms the high specificity. Based on the rarity of ASS, even small gaps in specificity (~99.0%) can result in more individuals with anti-OJ antibodies with a diagnosis different from ASS (or IIM) which can be attributed to the low pre-test probability due to vague clinical presentation. Since it is well appreciated that the results derived from LIAs lack reliability for certain analytes, different approaches have been proposed to enhance the clinical utility including adjusting the cut-off values [22] and combining the results with an IIF staining pattern [31,32]. A limitation of our study is that the final clinical diagnosis of the individuals tested by LIA was not obtained due to ethics constraints. However, the LIA samples represented real-world samples that were submitted for testing by physicians evaluating patients with suspected IIM.

Based on the observations summarized above, it has been speculated that the complex nature of the OJ antigen complex is the primary challenge in the cognate immunoassay development and performance [15]. However, recently it was demonstrated that two antigenic targets of the OJ complex (IARS and KARS) represent immunogenic antigens for the detection of anti-OJ antibodies [29]. More specifically, Muro et al. [29] demonstrated that an ELISA system using IARS and KARS detected 12/12 anti-OJ positive sera as identified by IP with 93.8% specificity. In contrast to Muro et al. [29], we observed a difference in the ability to detect anti-OJ positive samples. In the study by Muro, both IARS and KARS showed an area under the curve (AUC) under the curve of close to 1 indicating equal and high sensitivity for anti-OJ antibody detection. Our data indicates that the combination of the two antigens detects most samples identified by IP using a preliminary cut-off (of 100 MFI). After further assay optimization, validation studies are required to assess the possibility to use the antigens as a common approach to detect anti-OJ antibodies as viable alternative to IP. In addition to sensitivity studies, the specificity of the combination of the two antigens has to be assessed. In our study, we observed higher reactivity for anti-IARS in the cohort selected based on LIA and for anti-KARS in the cohort selected based on IP. This might be attributed to the use of IARS in the commercial LIA. Whether antibodies to KARS are more effectivein precipitating the OJ complex is speculative but should be considered for future studies.

An alternative approach for the detection of anti-synthetase antibodies was proposed by Aggarwal et al. [24] adopting IIF staining of HEp-2 cells ata reduced serum dilution. Using this approach, 2/5 anti-OJ positive samples demonstrated anti-cytoplasmic antibody staining. This approach requires validation using a larger cohort of anti-OJ sera that have been defined by consensus between IP and PMAT. 

### 4.2. Linear Peptides as Antigenic Target for Anti-OJ Antibodies

Although we observed reactivity to linear epitopes by anti-OJ positive sera, it is unlikely that linear peptides, like that demonstrated for other autoantibody systems such as PM/Scl [33], are sufficient to capture the entire B-cell immune response as a diagnostic approach to the detection of anti-OJ antibodies. Further studies are desired to further investigate reactivity to other OJ peptides. Along those lines, it is noteworthy that we used pooled sera to screen for B-Cell epitopes on OJ which might represent a limitation to find clues about multi-reactivity of anti-OJ antibody positive samples. However, this approach has been successfully used for other autoantibodies. The data on OJ peptides is in accord with the knowledge that both protein and RNA IP are more sensitive than LIA or other solid-phase immunoassays for anti-OJ antibody detection, which argues that conformational rather than linear epitopes are critical for autoantibody binding.

### 4.3. Anti-OJ Antibodies and Indirect Immunofluorescence

In contrast to other anti-ARS antibodies such as anti-Jo-1, anti-PL-7 or anti-PL-12, ICAP has not yet defined a specific pattern associated with anti-OJ antibodies. In addition to the general difficulty to find monospecific anti-OJ sera, a unique characteristic of anti-OJ samples that is different from other anti-ARS, is its macromolecular structure. Anti-OJ defined based on IP of the OJ complex consists of 10+ proteins and their cognate tRNA. Theoretically, reactivity with any component(s) of the OJ multiprotein-RNA complex can be considered anti-OJ positive. Thus, it is possible that IIF patterns of anti-OJ positive sera are heterogeneous reflecting the reactivity with different components, potentially even before they form the macromolecular complex. Perhaps, defining the IIF patterns of anti-IARS, EPRS, LARS, KARS (etc.) separately can provide a clearer answer. When we compared the reactivity of anti-OJ antibodies with IIF pattern, we could not identify a clear association with the anti-cellular (AC) patterns as defined by ICAP [34]. This was true for both LIA and PMAT and for two antigens, KARS and IARS. Unfortunately, no IIF data was available for samples selected based on IP and this should be addressed in future studies. 

**Table 2 diagnostics-13-00156-t002:** Overview of studies on anti-OJ antibodies and sensitivity of line immunoassay vs. immunoprecipitation.

Antibody	Sensitivity *	Comment
Tansley et al. [16]	0/14 (0%)	Poor sensitivity vs. IP
Mecoli et al. [22]	N/R	2/252 samples were positive for anti-OJ by LIA, but not confirmed by IP
Hamaguchi et al. [18]	0/9 (0%)	Poor sensitivity vs. IP
Lackner et al. [20]	N/R	All anti-OJ positive samples had diagnosis other than IIM
Vulsteke et al. [30]	N/R	0/144 IIM patients positive for anti-OJ
Cavazzana et al. [19]	0/2 (0%)	2/57 IIM patients had anti-OJ by IP, but did not confirm by LIA
Platteel et al. [35]	N/R	1/187 (0.5%) in IIM; 2/632 (0.3%) in controls; OR 0.2-18.5
Betteridge et al. [36]	N/R	10/1673 (0.6%) of IIM positive for anti-OJ antibodies
Espinosa-Ortega et al. [21]	0/1 (0%)	1/110 (0.9%) of IIM positive for anti-OJ antibodies

* Sensitivity of line immune assay vs. immunoprecipitation. Abbreviations: IIM, idiopathic inflammatory myopathies; IP, immunoprecipitation; LIA, line immunoassay; NR, not reported; OR, odds ratio.

## 5. Conclusions

Our data supportsthe findings reported by Muro et al. [29] that two components of the OJ complex (KARS and IARS) represent promising targets for the detection of anti-OJ antibodies. Consequently, the lack of accuracy of anti-OJ immunoassays to date is less likely linked to the complex nature of the OJ antigen, but rather to the selection of specific B cell targets and/or the strategy to develop the immunoassay for the detection. The data presented here has the potential to improve detection of anti-OJ antibodies.

## Figures and Tables

**Figure 1 diagnostics-13-00156-f001:**
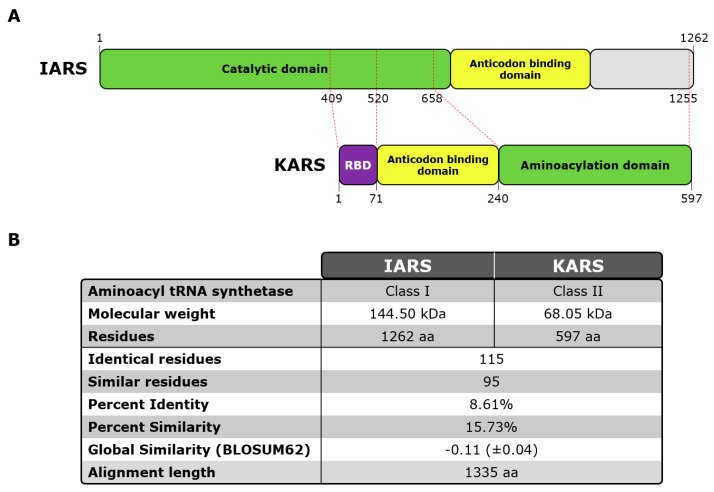
**KARS and IARS protein scheme and sequence alignment overview**. Legend: (**A**) Schematic view of human KARS and IARS proteins indicating functional domains reported in literature of each protein. Red dashed lines represent correlation (extracted from multiple sequence alignment) between KARS main domains and IARS amino acid sequence. (RBD: RNA Binding Domain). (**B**) Table containing protein sequence alignment of human KARS and IARS key features. Multiple alignment was carried out by Clustal Omega (RRID:SCR_001591). Abbreviations: aa, amino acids; kDa, kilodaltons; RBD, RNA binding domain.

**Figure 2 diagnostics-13-00156-f002:**
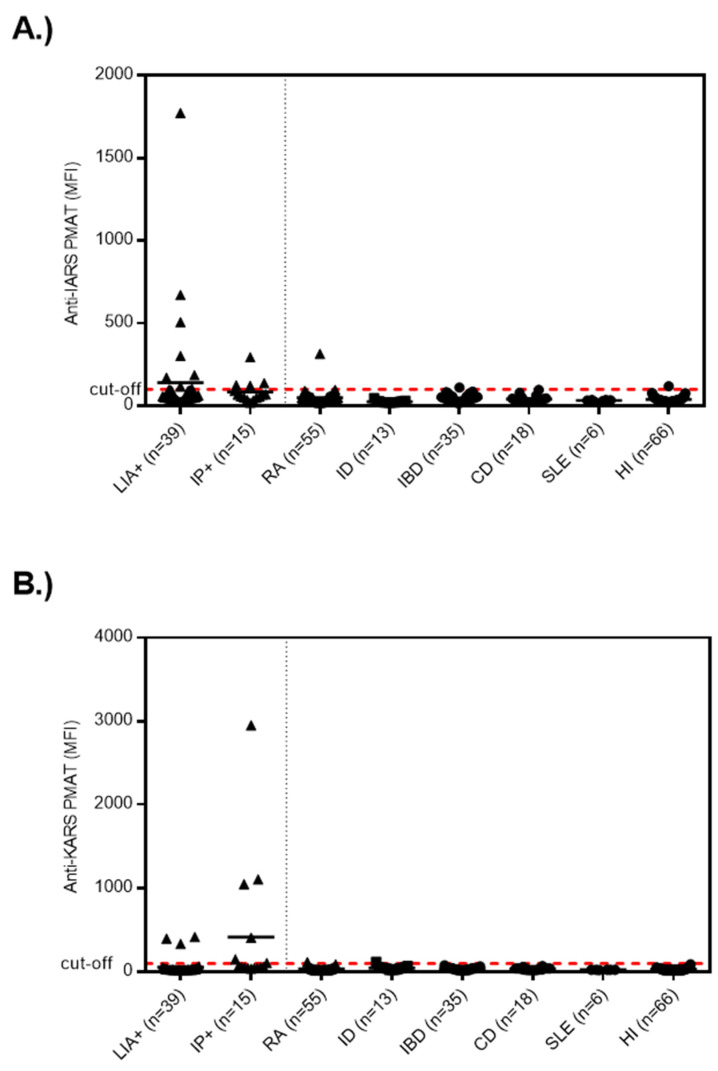
**Levels of anti-IARS and anti-KARS reactivity in samples selected either by line immunoassay (LIA) or immunoprecipitation (IP) and controls**. Legend: Samples selected based on LIA and IP had higher levels of anti-IARS and anti-KARS antibodies compared to controls. (**A**) exhibits the data for anti-IRAS and (**B**) for anti-KARS. Abbreviations: CD, celiac disease; LIA+, line immunoassay cohort; HI, healthy individuals; IBD, inflammatory bowel disease; ID, infectious disease; IP+, immunoprecipitation cohort; RA, rheumatoid arthritis; SLE, systemic lupus erythematosus.

**Figure 3 diagnostics-13-00156-f003:**
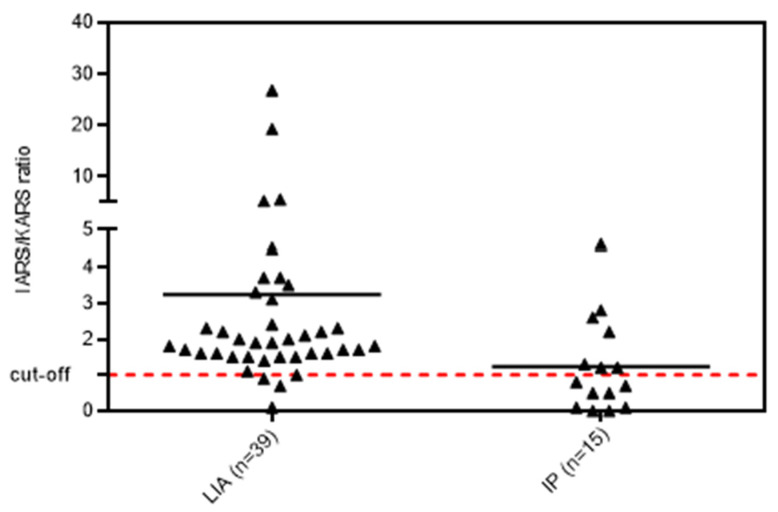
**Ratio between anti-IARS and anti-KARS reactivity in samples selected either by line immunoassay (LIA) or immunoprecipitation (IP). Cut-off represents equal reactivity to both antigens.** Legend: Samples that were selected based on LIA had significantly higher reactivity to IARS as compared to the KARS (*p* = 0.003). The cut-off dotted line indicates the threshold when both antigens (IRAS and KARS) showed the same reactivity.

**Figure 4 diagnostics-13-00156-f004:**
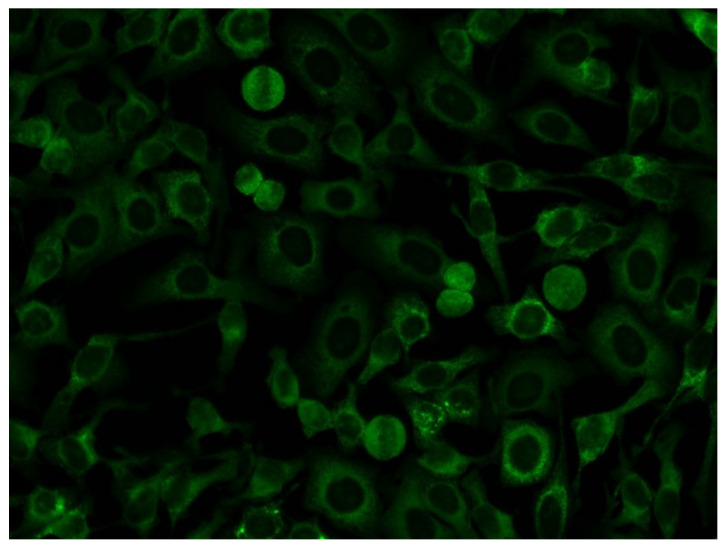
Example of the indirect immunofluorescence pattern (AC-19/20) on HEp-2 cells from a serum sample having high levels of anti-OJ antibodies identified by immunoprecipitation.

**Figure 5 diagnostics-13-00156-f005:**
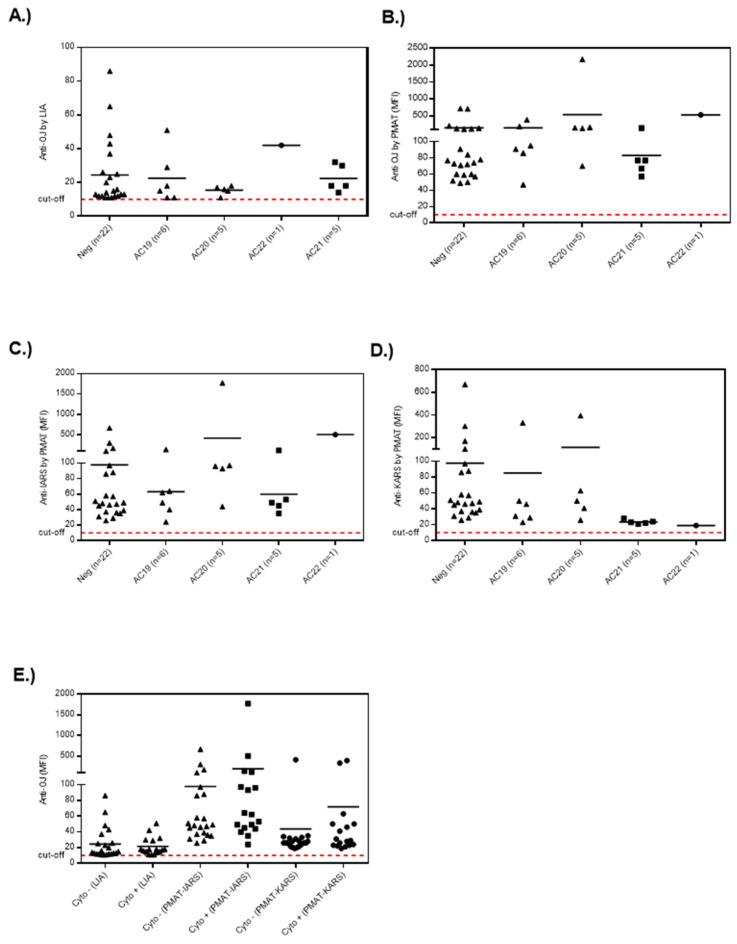
**Lack of association of anti-OJ antibodies with distinct indirect immunofluorescence (IIF) patterns**. Legend: Level of anti-OJ antibodies in relation to the IIF staining patterns (AC-19, cytoplasmic dense fine speckled; AC-20, cytoplasmic fine speckled; AC-21 anti-mitochondria-like; AC-22: Golgi complex) on HEp-2 cells for the results derived from particle based multi-analyte technology (PMAT) and line immunoassay (LIA). (**A**) Anti-OJ antibodies defined by LIA, (**B**), by PMAT (combination of IARS and KARS), (**C**) anti-IARS by PMAT, (**D**) anti-KARS by PMAT and (**E**) all assays separated by presence or absence of cytoplasmic staining.

**Figure 6 diagnostics-13-00156-f006:**
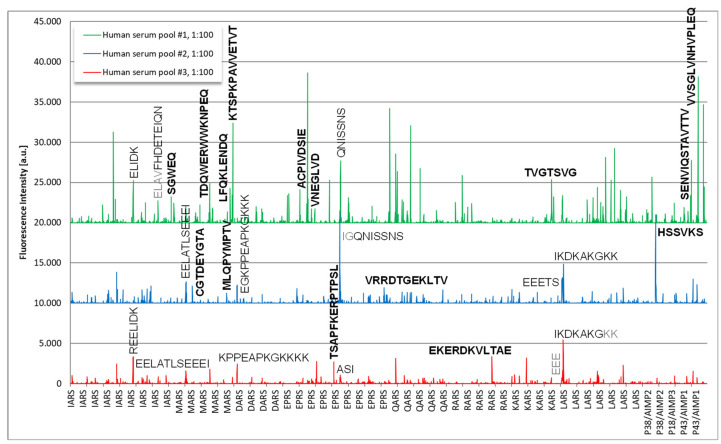
Epitope mapping of anti-OJ antibodies using solid phase peptide arrays. Legend: The sequences of antigens IARS, MARS, DARS, EPRS, QARS, RARS, KARS, LARS, P38/AIMP2, P18/AIMP3 and P43/AIMP1 were synthesized as 15mer amino acid peptides with a peptide to peptide overlap of 13 amino acids. Several immunoreactive peptides were found and their sequences are indicated in the figure. Major consensus sequences are shown in bold.

**Table 1 diagnostics-13-00156-t001:** Anti-OJ antibodies in samples selected based on a line immunoassay (LIA) or immunoprecipitation (IP) and measured by PMAT of IARS and KARS peptides.

Selection Method	IARS	KARS	Combined *
LIA (n = 39)	7/39 (17.9%)	3/39 (7.7%)	14/39 (35.9%)
IP (n = 15)	4/15 (26.7%)	6/15 (40.0%)	12/15 (80.0%)

* Note: the cutoff for IARS + KARS positives was lower than for the individual antigens. Hence, the number of positives is higher than the sum of the individual peptides.

## Data Availability

Data will be made available upon appropriate request.

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
