# Peer review of "Deciphering the Autoantibody Response to the OJ Antigenic Complex"

_diagnostics, 2023, doi:10.3390/diagnostics13010156_

Round 1

Reviewer 1 Report

This is a timely and interesting report that covers an important aspect of diagnosis for anti OJ positive myositis patients. The methods are robust, and extensively analysed and validated. The sample size is small, though thats a limitation by nature of the disease. The results are clinically meaningful and add value to existent literature. 

Author Response

Dear Reviewer,

We thank the reviewer for careful review of our manuscript. We hope that revised version of the paper is now suitable for publication in the journal of Diagnostics.  

Reviewer 2 Report

The paper investigates a novel approach to detect one species of myositis-specific auto-antibodies directed against the OJ antigen that are associated with IIM especially ILD. Currently used techniques have major pitfalls, either due to their low clinical performance (line immunoassay and dot blot) or due to the specific expertise required to perform the assays (in house developed ELISA, immunoprecipitation) that hampers their use in routine diagnostic laboratories. Since anti-OJ are directed against a macromolecular complex that comprises several tRNA transferases, the challenge is to find a suitable substrate for detection of the antibodies. Although anti-OJ reactivity is probably rare (<5% of IIM patients) and the disease can also be diagnosed without presence of autoantibodies, improving the diagnostic accuracy is a very relevant issue and the topic is therefore of interest to the field. The authors demonstrate that the use of full length KARS and IARS proteins in a particle-based multi-analyte technology might be a promising approach for the detection of anti-OJ. A consistent association with a nuclear or cytoplasmic IIF staining on HEp-2 cells was not observed in anti-OJ positive sera, neither in the total selection of 39 LIA positive samples nor in the LIA positive samples that also tested positive in PMAT. The paper is generally well written but can be improved.

Major comments:

1)   The authors have investigated anti-OJ reactivity by fully automated particle-based multi-analyte technology (PMAT). Two antigens from the macromolecular complex were chosen, KRAS and IRAS, based on a recent publication in which these two antigens were also used in an ELISA. The authors conclude that “the combination of IARS and KARS might represent a promising approach for the detection of anti-OJ antibodies on a fully automated platform”. The choice for only two antigens should be discussed, since there are many more potential antigens in the OJ complex and anti-OJ positive sera show high reactivity to intact antigens as shown in IP (see ref 29) but also to linear epitopes as demonstrated by the authors in figure 5.

2)   The authors summarize extensively previous data for detection of anti-OJ with LIA and IP and included a table in addition to the text (table 2). Since the paper is not a review, these data should be substantially shortened and be presented in the introduction.

3)   It is not clear how the 39 LIA positive sera were selected. From the 39 selected sera, only 11 showed a positive to strong positive reaction (>25) and all others were weak positive. It is known that weak positive LIA results have a low specificity.

Together with the fact that no clinical data were available other than that the test was ordered because of a clinical suspicion of IIM  it is therefore not possible to draw conclusions on the performance of the newly developed PMAT compared to the LIA.

Minor comments:

1)    Line 127: How are the anti-OJ positive pool sera used for epitope mapping defined? Please discuss why pooled sera have been used instead of individual sera. Would the epitope mapping with individual sera give more clues to the potential multi-reactivity of anti-OJ positive patients?

2)    Line 151: It is stated that 6 synthetic soluble biotinylated peptides representing key epitopes were used. I could not find corresponding data in the paper. The authors discuss that although linear peptides are recognized by anti-OJ positive sera as shown in figure 5, it is unlikely that they are sufficient to capture the entire anti-OJ response and conformational rather than linear epitopes are critical for autoantibody binding. Thus, use of peptide antigens is of no use in a diagnostic approach?

3)    Line 79: Results presented in paragraph 3.1 are difficult to interpret. It is not clear from the text whether results are presented for reactivity to full recombinant proteins or to peptides since the terms KARS and IARS antigen and peptides are both used in the text without further explanation. This should be clarified.

4)    Line 192: How is the combined cut-off for anti-KARS and anti-IARS defined?

5)    Line 236: Figure 3: The authors state that the large cytoplasmic dot pattern that shows the highest fluorescence intensity is some cells is not likely related to anti-OJ. I would conclude that the picture is therefore not representative and should be replaced by another example.

Author Response

Dear reviewer,

We thank the reviewers for careful review of our manuscript. We strongly believe that the modifications following the advice from the reviewers improved the manuscript. We hope that revised version of the paper is now suitable for publication in the journal of Diagnostics.  
